# DiffSparse: Accelerating Diffusion Transformers with Learned Token Sparsity

**Haowei Zhu**[1,2]**, Ji Liu**[1]**, Ziqiong Liu**[1]**, Dong Li**[1]**, Junhai Yong**[2]**, Bin Wang**[2,3]***, Emad Barsoum**[1]

[1] Advanced Micro Devices, Inc. [2] Tsinghua University [3] BNRist
{haoweiz, liuji, ziqioliu, d.li, ebarsoum}@amd.com,
yongjh@tsinghua.edu.cn, wangbins@tsinghua.edu.cn

## Abstract

Diffusion models demonstrate outstanding performance in image generation, but their multi-step inference mechanism requires immense computational cost. Previous works accelerate inference by leveraging layer or token cache techniques to reduce computational cost. However, these methods fail to achieve superior acceleration performance in few-step diffusion transformer models due to inefficient feature caching strategies, manually designed sparsity allocation, and the practice of retaining complete forward computations in several steps in these token cache methods. To tackle these challenges, we propose a differentiable layer-wise sparsity optimization framework for diffusion transformer models, leveraging token caching to reduce token computation costs and enhance acceleration. Our method optimizes layer-wise sparsity allocation in an end-to-end manner through a learnable network combined with a dynamic programming solver. Additionally, our proposed two-stage training strategy eliminates the need for full-step processing in existing methods, further improving efficiency. We conducted extensive experiments on a range of diffusion-transformer models, including DiT-XL/2, PixArt-$\alpha$, FLUX, and Wan2.1. Across these architectures, our method consistently improves efficiency without degrading sample quality. For example, on PixArt-$\alpha$ with 20 sampling steps, we reduce computational cost by $54\%$ while achieving generation metrics that surpass those of the original model, substantially outperforming prior approaches. These results demonstrate that our method delivers large efficiency gains while often improving generation quality.

## 1 Introduction

In recent years, diffusion models have made remarkable progress in the field of image generation. Among them, the Stable Diffusion series (Rombach et al., 2022; Podell et al., 2023; Tian et al., 2024; Esser et al., 2024) has achieved significant success in high-quality image generation (Zhu et al., 2024; 2025a). This advancement is largely attributed to the effectiveness of diffusion probabilistic models (DPM) (Ho et al., 2020) and the powerful U-Net (Ronneberger et al., 2015) architecture, which allows high resolution synthesis with detail preservation. Additionally, some recent works (Peebles & Xie, 2023b; Chen et al., 2024b; Tian et al., 2024) have explored the integration of diffusion models with Transformer-based architectures, demonstrating outstanding performance. In particular, scaling laws have been leveraged to expand the model size of Transformers (Vaswani et al., 2017), further enhancing precision and generative quality, these large-scale models leverage improved expressivity and enhanced generalization, pushing the boundaries of generative artificial intelligence.

However, despite these advances, the substantial computational cost associated with diffusion models presents a significant challenge for real-world deployment. The inference of such large models requires extensive computational resources, which can hinder practical applications. Addressing this issue requires innovations in model acceleration techniques to enable broader accessibility and usability of diffusion-based generative models. Existing methods of diffusion model acceleration typically focus on sampler optimization (Song et al., 2020; Lu et al., 2022a), model pruning (Fang et al., 2023b; Zhang et al., 2024; Fang et al., 2023a), distillation (Yin et al., 2024b; Luo et al., 2023;

---

*Corresponding author.

Salimans & Ho, 2022), and feature caching (Selvaraju et al., 2024; Ma et al., 2024; Liu et al., 2025a;b). Feature caching methods leverages temporal redundancy to reuse intermediate features, achieving significant speedups. They become popular in the field of diffusion model acceleration due to non-training diffusion model and easy integrating into the original inference pipeline. Previous methods cache and reuse coarse-grained, layer-level features, whereas token cache methods (Zou et al., 2025; 2024; Zhang et al., 2025) reuse token-level features, achieving better acceleration performance. However, these approaches require manual sparsity allocations and hand-crafted schedules that preserve several full forward passes during denoising, which limits the acceleration potential of token-level feature caching.

To address these challenges, we propose DiffSparse, a learnable framework for optimizing layer-wise sparsity allocation in diffusion transformer models. Our approach dynamically determines the optimal sparsity configuration across all layers and inference steps, ensuring that the overall pruning rate is met in an end-to-end manner through a model-driven process. Moreover, DiffSparse eliminates the need for complete forward computations in predefined steps required by existing methods, further enhancing efficiency.

Specially, our approach formulates the token cache optimization as a dynamic programming-based sparsity allocation problem. We innovatively design a learnable sparsity cost predictor, which predicts a cost matrix that quantifies the sparsity costs associated with target sparsity rates for all layers across every denoising step. Then we propose a dynamic programming approach to determine the optimal sparsity configurations for all layers over the relevant denoising steps, minimizing the overall sparsity cost while satisfying the required sparsity rate. Finally, we introduce a token selector that dynamically selects a specific proportion of tokens for reuse, leveraging the learned sparsity ratio to accelerate inference. To optimize the learnable sparsity cost predictor, we utilize a perceptual distillation loss that minimizes the degradation in generation quality. Furthermore, we introduce a two-stage training strategy that eliminates the need for complete forward computations in predefined steps required by existing methods while also improving accuracy. We have conducted extensive experiments on various transformer-based baselines, and the pruning results outperform other SOTA pruning methods by a large margin. For example, pruning $54\%$ of tokens yields an FID of 27.79, our method substantially better than the state-of-the-art methods ToCa (28.35) and TaylorSeer (29.08), while achieving a higher speedup ($1.91\times$) on PixArt-$\alpha$. These results underscore the practical effectiveness of our method. Our contributions are summarized as follows:

- We propose DiffSparse, a differentiable approach to optimize layer-wise token sparsity in diffusion models sampling process. By integrating a sparsity cost predictor, dynamic programming solver, and adaptive token selector, it automates sparsity allocation and token reuse without manual heuristics.

- We introduce a two-stage training strategy that eliminates the need for predefined complete forward computations in several steps required by existing methods, fully unlocked the acceleration potential of token-level feature caching.

- Extensive experiments on diverse foundation models prove that our method surpasses existing SOTA methods by a large margin, setting new efficiency-accuracy benchmarks.

## 2  RELATED WORK

**Diffusion Transformer Models.** The integration of transformers into diffusion models has significantly advanced generative modeling, improving scalability and performance. Diffusion models, which generate data by iteratively denoising from a noise distribution. Traditionally, diffusion models relied on CNNs, but recent studies demonstrate the effectiveness of transformers (Peebles & Xie, 2023b; Chen et al., 2024b; Tian et al., 2024; Brooks et al., 2024; Chen et al., 2024a; Wu et al., 2025). Diffusion Transformer (DiT) (Peebles & Xie, 2023b) replaces the U-Net backbone with a transformer, leveraging long-range dependencies and efficient scaling to achieve superior image generation. PixArt (Chen et al., 2024b) builds on this by introducing a hierarchical transformer architecture and a novel noise schedule, excelling in high-resolution and text-to-image synthesis. Although diffusion transformer models have achieved great success, the substantial computational overhead from the iterative denoising process makes them inefficient for industrial deployment.

**Acceleration of Diffusion Models.** Diffusion acceleration is a critical research area focused on reducing computational costs and improving inference efficiency while preserving high-quality generation. Recent advancements can be categorized into sampler optimization (Song et al., 2021; Lu et al., 2022a;b), model pruning (Fang et al., 2023b; Zhang et al., 2024), distillation (Salimans & Ho, 2022; Yin et al., 2024a), and feature caching (Li et al., 2023; Ma et al., 2024; Zhu et al., 2025b). Sampler optimization reduces the number of denoising steps during inference using deterministic or adaptive strategies to approximate the denoising process efficiently. Model pruning removes redundant parameters and achieving speedups with structured pruning (Fang et al., 2023a). Other strategies, such as Rectified Flow (Liu et al., 2022) and knowledge distillation (Yin et al., 2024a) accelerates inference by matching model outputs in fewer steps without quality loss.

Feature caching is particularly effective for DiT architectures. Methods such as FORA (Selvaraju et al., 2024) and $\Delta$-DiT Chen et al. (2024c) reuse attention and MLP representations, while DiTFastAttn (Yuan et al., 2024) further reduces redundancies in self-attention. Dynamic strategies like TeaCache (Liu et al., 2025a) estimate timestep-dependent differences, and TaylorSeer (Liu et al., 2025b) introduced a "cache-then-forecast" paradigm that predicts and updates cached features, though its advantage is most evident with long-range caching. SpeCa (Liu et al., 2025c) further enhance the performance with speculative sampling. Complementary to these are token cache methods (Zou et al., 2025; 2024; Zhang et al., 2025; You et al., 2025), which apply fine-grained, error-guided token-wise caching to dynamically update features, achieving substantial acceleration without compromising quality. More discussion with existing methods are presented in Appendix.

In this paper, we introduce DiffSparse, a feature-caching approach for accelerating diffusion transformer models. These models typically require only a few dozen sampling steps and have seen growing adoption in industry. Unlike prior works (Zou et al., 2025; 2024), DiffSparse employs a token-level cache within an end-to-end learning framework that casts model acceleration under a fixed compression ratio as a layer-wise sparsity optimization problem across timesteps, eliminating the need for manually tuned sparsity or acceleration parameters. To address inefficiencies in existing approaches, which depend on predefined full-step computation schedules, we also propose a two-stage training protocol that adaptively allocates computation where it is most needed.

## 3 METHOD

In this section, we start with a brief introduction to the diffusion transformer model and the token cache strategy. We then present the challenges of the existing token caching approaches. Finally, we present our DiffSparse approach, which builds upon the token cache strategy for acceleration and optimizes the layer-wise token sparsity of diffusion transformer model in a learnable manner, enhancing accuracy while maintaining the sparsity requirement.

### 3.1 PRELIMINARY

**Diffusion Models.** Diffusion models are a class of generative models that construct a Markov chain of latent variables by progressively adding Gaussian noise to data samples and then reversing this process to synthesize new samples. Given an initial data sample $x_0$, the forward diffusion process transforms the data through a series of steps:

$$q(x_t \mid x_{t-1}) = \mathcal{N}\left(x_t; \sqrt{1 - \beta_t}\, x_{t-1},\, \beta_t \mathbf{I}\right), \tag{1}$$

where $t$ is the time step, $\{\beta_t\}_{t=1}^T$ denotes a predefined variance schedule. After $T$ steps, the data is nearly transformed into an isotropic Gaussian distribution, i.e., $q(x_T) \approx \mathcal{N}(0, \mathbf{I})$.

The reverse process is parameterized by a noise prediction network, which aims to recover the original data by iteratively removing the added noise, and is modeled as:

$$p_\theta(x_{t-1} \mid x_t) = \mathcal{N}\left(x_{t-1}; \mu_\theta(x_t, t),\, \Sigma_\theta(x_t, t)\right), \tag{2}$$

where $\mu_\theta$ and $\Sigma_\theta$ are learned functions. Because the network is applied at each timestep in the multi-step denoising process, the repeated evaluations of the noise prediction network dominate the computational cost, accounting for the majority of the model's floating-point operations (FLOPs).

**Diffusion Transformer.** The Diffusion Transformer (Chen et al., 2024b) is a novel architecture that synergizes the iterative refinement capabilities of diffusion processes with the representational power of transformers. In this framework, the input is represented as a set of tokens $\mathbf{X} \in \mathbb{R}^{N \times D}$, where $N$ denotes the number of tokens and $D$ their dimensionality. The network architecture is composed of $L$ stacked blocks, each integrating three key components: a self-attention (SA) layer, a cross-attention (CA) layer, and a multi-layer perceptron (MLP) layer. The self-attention mechanism enables the model to capture long-range dependencies among tokens. In parallel, the cross-attention module facilitates the incorporation of conditioning information, enhancing the model's ability to generate contextually relevant outputs. The subsequent MLP further refines these token representations through non-linear transformations.

A significant advantage of the Diffusion Transformer lies in its ability to iteratively refine token representations during the denoising process, leading to improved sample quality. This layered approach allows the model to effectively balance global context and local details, thereby offering enhanced performance in complex generative tasks.

**Token-Wise Feature Caching Approach.** Prior work (Ma et al., 2024) has demonstrated that features at adjacent timesteps exhibit high similarity, leading to significant redundancy. To exploit this redundancy for computational efficiency, previous approaches (Ma et al., 2024; Wimbauer et al., 2024) have introduced caching mechanisms that reuse features to accelerate processing. The token-wise feature caching approach (Zou et al., 2025) operates at a finer granularity by caching features at the individual token level, enabling more effective exploitation of the redundancy.

Token-wise feature caching mechanism begins by computing and storing the intermediate token features $\mathbf{X} = \{\hat{x}_0, \hat{x}_1, \ldots, \hat{x}_{N-1}\}$ from each self-attention, cross-attention, and MLP layer into a cache $C$ at the initial timestep $t$. In subsequent timesteps, a predefined cache ratio $R$ determines the proportion of tokens reused from the cache $C$ for each layer at each timestep. The $R$ selected tokens based on token importance rank, denoted as $I_{\text{Cache}}$, will bypass re-computation by reusing their cached values, while the remaining tokens $I_{\text{Compute}} = \{\hat{x}_i\}_{i=1}^{N} \setminus I_{\text{Cache}}$ are recomputed. For a given layer $f$, the computation for each token $\hat{x}_i$ is formulated as:

$$F(\hat{x}_i) = \gamma_i f(\hat{x}_i) + (1 - \gamma_i)C(\hat{x}_i), \tag{3}$$

where $\gamma_i = 0$ for $\hat{x}_i \in I_{\text{Cache}}$ and $\gamma_i = 1$ for $\hat{x}_i \in I_{\text{Compute}}$. To mitigate error accumulation from reused features, the cache is dynamically updated for tokens in $I_{\text{Compute}}$ via:

$$C(\hat{x}_i) \leftarrow F(\hat{x}_i). \tag{4}$$

This token-wise feature caching approach effectively reduces redundant computations by leveraging the high similarity of features across adjacent timesteps, thus significantly accelerating the inference process while maintaining robust feature representations.

**Challenges in Existing Token Caching Approaches.** While token caching methods (Zou et al., 2025) have shown great promise in speeding up diffusion transformers, key limitations remain. First, they require manually setting a reuse sparsity rate for each layer at every timestep, resulting in a large, hard-to-tune parameter space. This manual process hampers performance and scalability. A learnable or adaptive sparsity strategy could unlock further gains. Second, current methods still depend on a full-step design (several steps without caching) to maintain generation quality. However, this compromises the efficiency of token-based operations. Replacing this with dynamic caching tailored to diffusion transformers can better balance quality and speed. In this paper, we propose an intelligent framework that jointly learns optimal sparsity across layers and removes the reliance on full-step computation, significantly improving both performance and flexibility.

## 3.2 DIFFSPARSE APPROACH

To automate per-layer sparsity selection and remove the reliance on full-step designs, we propose DiffSparse, an efficient token caching framework for diffusion transformers. DiffSparse learns layer-wise sparsity end-to-end by combining a learnable sparsity cost predictor with a dynamic programming solver to find optimal sparsity configurations across layers and denoising steps. It also adopts a two-stage training scheme that gradually replaces full computation steps with cache-based ones, improving efficiency without sacrificing performance. As illustrated in Figure 1, DiffSparse

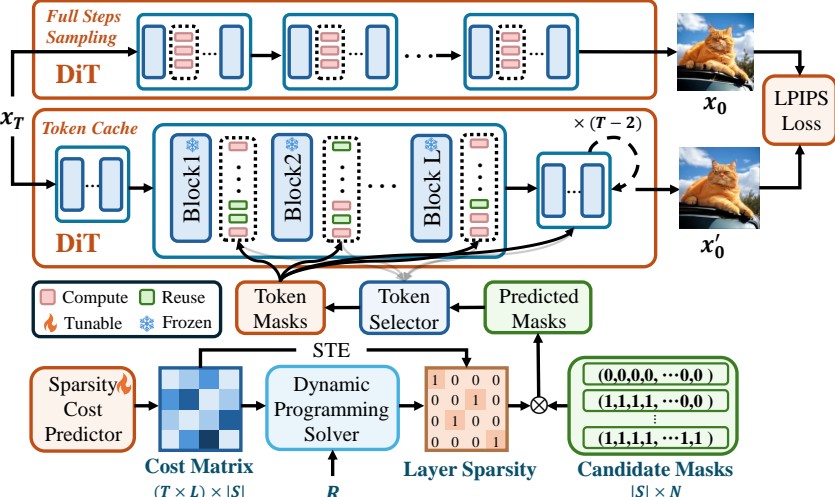

Figure 1: DiffSparse uses a learnable sparsity-cost predictor and dynamic programming to learn per-layer sparsity under target ratio $R$. We generate binary masks from the chosen sparsity maps and candidate masks. A token selector reuses features from previous diffusion steps to skip unimportant tokens and speed sampling. To enable gradient flow through the binary masks, we apply Straight-Through Estimation (STE) and train our model using full-step sampling targets with LPIPS loss.

comprises three components: a token selector, a sparsity cost predictor, and a dynamic programming solver. The cost predictor estimates a cost matrix representing the sparsity cost for various predefined rates across all layers and denoising steps (excluding the first). The dynamic solver then identifies the optimal sparsity pattern under a global sparsity constraint $R$. Based on this, the token selector determines which tokens to reuse and which to recompute at each layer. Training is guided by a perceptual distillation loss, integrated into a two-stage training pipeline for effective learning.

**Token Selector.** We employ a *Token Selector* that assigns each token $\hat{x}_i$ an importance score used to decide which tokens are freshly computed and which remain cached. The score is a composite, layer-wise quantity of the form:

$$S(\hat{x}_i) \;=\; B\Big( \sum_{q=1}^{Q} \lambda_q \, s_q(\hat{x}_i) \Big), \tag{5}$$

where each $s_q(\hat{x}_i)$ is a scalar signal capturing a different criterion (for example, self-attention influence, cross-attention focus, cache-reuse frequency, *etc.*), and $\{\lambda_q\}_{q=1}^{Q}$ are weighting hyperparameters that balance these criteria. The operator $\mathcal{B}(\cdot)$ is optional and denotes a spatial bonus operation that promotes a spatially uniform coverage of selected tokens (implemented, e.g., by boosting tokens that are local maxima within a $k \times k$ neighborhood). Other choices for $\mathcal{B}$ are possible (e.g. smooth kernels or distance-based adjustments).

Given the per-token scores $S(\hat{x}_i)$ in a layer with $N$ tokens, we sort tokens by descending score and select the top $K$ tokens according to a predefined sparsity ratio $R$. We emphasize that our contribution is orthogonal to any particular token-ranking heuristic: the choice of scoring components (e.g. self-attention influence, cross-attention terms, spatial bonus) is optional and can be replaced by alternative ranking methods. Detailed descriptions and comparisons of specific token-ranking strategies are provided in the Appendix A.5.1. Empirically, our allocation scheme yields consistent gains across different token-ranking methods (see Table 5).

**Learnable Sparsity Cost Predictor.** We propose a learnable sparsity cost predictor to adaptively determine layer-wise sparsity in diffusion transformers (DiTs) while balancing inference efficiency and computational cost. Given a DiT with $L$ layers operating over $T$ denoising timesteps, our goal is to generate a binary mask $M \in \{0,1\}^N$ for each layer $l$ and timestep $t$ that selects $K_{l,t}$ tokens

for full computation and reuses features for the remaining $N - K_{l,t}$ tokens. This is formalized as a constrained optimization over a candidate sparsity set $S$, where $|S|$ denotes the number of predefined sparsity configurations. For a layer containing $N$ tokens, let $S$ denote the set of sparsity rates, each a value between 0 and 1, at which we retain a corresponding fraction of tokens. For instance, if $N = 256$ and we choose a step size of 32 tokens, we obtain $S = \{0, 0.25, 0.50, 0.75, 1.0\}$, which corresponds to retaining $\{0, 64, 128, 192, 256\}$ tokens, respectively. Our objective is to learn the relative cost of applying different sparsity rates across layers. Our experimental results (Table 4) demonstrate that the learned sparsity predictor generalizes across resolutions, so that a sparsity allocation trained at low resolution remains effective at higher resolutions.

We implement the sparsity cost predictor using $(T \times L) \times |S|$ learnable parameters, where $T$ is the number of timesteps, $L$ is the number of layers, and $|S|$ is the size of the candidate sparsity set. The predictor outputs a normalized cost matrix $C \in \mathbb{R}^{(T \times L) \times |S|}$, where each entry $C_{(t,l),s}$ quantifies the cost of applying sparsity configuration $s \in S$ to layer $l$ at timestep $t$. We minimize the cumulative cost while ensuring the total sparsity meets a predefined overall pruning rate $R$. The sorted token set $\bar{\mathbf{X}} \in \mathbb{R}^{N \times D}$ enables efficient mask selection by prioritizing tokens with high scores.

Importantly, The cost predictor's size depends only on $T$, $L$, and $|S|$, not on token-sequence length $N$. Empirically, we found that simply increasing $|S|$ beyond a moderate size yields diminishing or negative returns (Table 7), and experiments show the learned cost predictor transfers across resolutions (Table 4), demonstrating scalability to high resolutions and robustness to token-length variation.

**Dynamic Programming Solver.** To determine the optimal sparsity configuration while satisfying a global sparsity constraint, we employ a dynamic programming approach to minimize the overall cost across layers. Formally, we define the state function:

$$F(\hat{l}, r) = \min_{\{s_i\}_{i=1}^{\hat{l}}} \sum_{i=1}^{\hat{l}} C_{i,s_i}, \quad \text{s.t.} \quad \sum_{i=1}^{\hat{l}} s_i = r, \tag{6}$$

where $F(\hat{l}, r)$ represents the minimum achievable cost when assigning sparsity levels to the first $\hat{l}$ layers under a total sparsity constraint $r$. The recursive formulation is given by:

$$F(\hat{l}, r) = \min_{s \in S, s \leq r} \left( F(\hat{l} - 1, r - s) + C_{\hat{l},s} \right). \tag{7}$$

Here, the transition considers all possible sparsity levels $s$ that can be allocated to layer $\hat{l}$, ensuring that the total sparsity constraint is maintained. The algorithm iteratively computes $F(\hat{l}, r)$ for $\hat{l} = 1, \ldots, L \cdot T$ and $r = 0, \ldots, \hat{R}$, followed by a backtracking step to reconstruct the optimal sparsity allocation, where $\hat{R} = R \cdot L \cdot T$. This approach operates with a time complexity of $O((L \cdot T)^2 \cdot |S|)$, making it computationally feasible for practical deep learning scenarios. To reduce the number of redundant state computations and lower overall complexity, we implement pre-pruning strategies. For example, when target sparsity ratio $R = 43\%$, $|S| = 5$, $T = 20$, and $L = 28$, it requires about 4 hours (including DP optimization and fine-tuning) of total training time. The DP solver runs in approximately $\approx$30 seconds for the configurations reported, but it is not executed at inference time. At inference, the model only uses the precomputed masks. Since the direct conversion of the predicted cost matrix $C$ to a discrete mask $M$ is non-differentiable, we utilize the Straight-Through Estimator (STE) (Jang et al., 2016) to approximate the gradients of the discrete mask with respect to the cost predictions. This approach facilitates end-to-end optimization of the sparsity cost predictor.

**Training Loss.** To guide the optimization of the pruned Diffusion Transformer, we employ the Learned Perceptual Image Patch Similarity (LPIPS) loss (Zhang et al., 2018) as a perceptual distillation loss. In our framework, the original model prior to token pruning serves as the teacher network, while the pruned model is treated as the student network. Both models generate outputs via a multi-step sampling process inherent to diffusion models.

Let $x_0$ and $x_0'$ denote the multi-step sampling outputs from the teacher and student networks, respectively. The LPIPS loss is then defined as:

$$\mathcal{L}_{\text{LPIPS}} = \text{LPIPS}(x_0, x_0'), \tag{8}$$

which measures the perceptual similarity between the outputs. During training, gradients are back-propagated solely through the student network, as the teacher network's parameters are detached (i.e.,

its gradients are not computed). This setup ensures that the student model is effectively distilled to mimic the perceptual characteristics of the teacher model, thereby achieving acceleration through token pruning while preserving output quality.

**Two-Stage Training Strategy.** We propose a two-stage training framework to optimize the cost matrices for `full-step positions` and `layer sparsity` components. In the first stage, we follow (Selvaraju et al., 2024; Zou et al., 2025) to preset $T_f$ full-step positions and independently optimize the step cost matrix $C_f \in \mathbb{R}^{T \times 2}$ encoding temporal sparsity decisions and the layer sparsity cost matrix $C_l \in \mathbb{R}^{(L \times T) \times |S|}$ governing token retention per layer. We first solve $C_f$ via dynamic programming to identify $|T_f|$ optimal full-step positions with minimal cumulative cost. For these selected steps, we warm-start layer sparsity optimization by subtracting $\delta$ from the predicted costs:

$$C_l^{(t,l,s)} \leftarrow C_l^{(t,l,s)} - \delta \quad \forall t \in T_f, l \in \{1, ..., L\}, s = N. \tag{9}$$

This strategy preserves inter-layer cost ranking while leveraging full-step error correction capabilities.

In the second stage, we integrate step and layer costs by modifying layer sparsity entries using Equation 9. The unified cost matrix is then fine-tuned to systematically redistribute FLOPs across sampling steps. Unlike existing methods (Selvaraju et al., 2024; Zou et al., 2025) that rigidly enforce full steps for noise correction, our approach dynamically optimizes sparsity patterns through differentiable cost interaction. The pseudocode is provided in the supplementary materials.

# 4 EXPERIMENTS

## 4.1 EXPERIMENT SETTINGS

**Model Configurations.** We conduct experiments on four widely used DiT-based models across various generation tasks: (1) **PixArt-$\alpha$** with 20 DPM Solver++ (Lu et al., 2022b) steps and **FLUX.1-schnell** (Labs, 2024) with 4 steps for text-to-image generation and (2) **DiT-XL/2** with 50 DDIM (Song et al., 2021) steps for class-conditional image generation. (3) **Wan2.1-1.3B** (Wan et al., 2025) with 25 flow-matching sampling steps for text-to-video generation. We define the candidate set $S$ as the range from 0 to 1 with an interval of 0.25, yielding $|S| = 5$ token sparsity candidates. More details of the implementation are provided in the supplementary material.

**Training.** For PixArt-$\alpha$ (Chen et al., 2024b) and FLUX.1-schnell (Labs, 2024), we train the learnable sparsity-cost predictor on 10,000 captions randomly sampled from the COCO (Lin et al., 2014) train dataset. For DiT-XL/2 (Peebles & Xie, 2023a), we use 10,000 ImageNet (Deng et al., 2009) train category indices, and for Wan2.1 we sample 10,000 captions from WebVid-10M (Bain et al., 2021) for training. During training we use no image data, only captions or class-conditioning information, which do not overlap with the evaluation set.

We leverage the layer sparsity configuration in the token-cache-based model (Zou et al., 2025) to initialize our sparse router training. All the models are trained with AdamW optimizer. The sparsity cost predictor is trained in two stages. For the first stage, the layer sparsity cost component is optimized for 1 epoch with a learning rate of $\eta = 1.0$, while the step cost component is trained separately using $\eta = 0.01$ to capture temporal patterns across denoising steps. For the second stage, we integrate the step cost into the layer-wise costs with $\delta = 10$ and then fine-tuned for 1 epoch with $\eta = 0.1$ to optimize layer sparsity allocation. Training requires approximately 4-10 hours on 8 AMD MI250 GPUs with 80GB memory per experiment.

**Evaluation.** For text-to-image generation, we evaluate on the **COCO** dataset (Lin et al., 2014) using 30,000 samples at $256 \times 256$ resolution and **PartiPrompts** (Yu et al., 2022) with 1,632 samples. Image quality is quantified by **FID-30k** (Heusel et al., 2017), which compares generated images against originals, while text-image alignment is measured by two complementary metrics: **CLIP-Score** (computed with CLIP-ViT-Large-14 (Hessel et al., 2021)) and **Image Reward** (Xu et al., 2023), a metric shown to more accurately reflect human preferences. For class-conditional image generation, 50,000 images at $256 \times 256$ resolution are generated from 1,000 **ImageNet** (Deng et al., 2009) classes and evaluated using the **FID-50k** metric. We evaluate text-to-video generation using the **VBench** framework on 950 prompts, generating 4,750 videos at $256 \times 256$ resolution, each lasting 2 seconds at 8 frames per second, and assess them across **16 metrics**.

## 4.2 MAIN RESULTS

**Results on Text-to-Image Generation.** We compare DiffSparse with existing methods under identical sparsity budgets. FORA, DeepCache (CVPR'24) and TaylorSeer (ICCV'25) are evaluated with cache interval $N = 2$, while DiCache, ToCa (ICLR'25) and DuCa are tested using their respective optimal configurations. Table 1 shows that DiffSparse delivers both faster inference and improved generation quality compared with existing methods. At roughly $1.74\times$ speed-up, existing methods suffer degraded image quality, while DiffSparse achieves a strong FID of 26.91 (vs. TaylorSeer's 29.08 and ToCa's 28.35). This corresponds to a relative +5.1% improvement in FID of DiffSparse over ToCa. Pushing further, DiffSparse attains $1.91\times$ acceleration while producing an FID that surpasses the original (full) model. This improvement stems from a learned sparsity schedule that accelerates convergence of the generated image distribution and improves visual fidelity, while preserving semantic alignment with the conditioning signal. We provide additional text-to-image comparisons in Appendix A.6, and also present more qualitative visual comparison in Appendix A.7.

Table 1: Results of text-to-image generation on MS-COCO2017 with PixArt-$\alpha$ and 20 DPM++ steps.

| Method | MACs (T)↓ | Speedup↑ | FID-30k↓ | CLIP↑ |
|---|---|---|---|---|
| PixArt-$\alpha$ (Chen et al., 2024b) | 2.86 | $1.00\times$ | 28.20 | 0.163 |
| 50% steps | 1.43 | $1.74\times$ | 37.57 | 0.158 |
| FORA ($\mathcal{N} = 2$) (Selvaraju et al., 2024) | 1.43 | $1.64\times$ | 29.67 | 0.164 |
| DeepCache ($\mathcal{N} = 2$) (Ma et al., 2024) | 1.48 | $1.61\times$ | 29.61 | 0.163 |
| DiCache (Bu et al., 2025) | 1.63 | $1.77\times$ | 28.19 | 0.164 |
| ToCa (Zou et al., 2025) | 1.64 | $1.75\times$ | 28.35 | 0.164 |
| DuCa (Zou et al., 2024) | 1.63 | $1.78\times$ | 27.98 | 0.164 |
| TaylorSeer (Liu et al., 2025b) | 1.57 | $1.83\times$ | 29.08 | 0.163 |
| **DiffSparse (R = 43%)** | 1.64 | $1.74\times$ | **26.91** | **0.164** |
| **DiffSparse (R = 54%)** | **1.30** | **$1.91\times$** | 27.79 | 0.164 |

Table 2: Results of class-conditional generation with DiT-XL/2 and 50 DDIM steps on ImageNet.

| Method | MACs (T) ↓ | Speedup↑ | FID↓ | sFID ↓ | Precision ↑ | Recall ↑ |
|---|---|---|---|---|---|---|
| DDIM-50 steps | 11.44 | $1.00\times$ | 2.26 | 4.29 | 0.80 | 0.60 |
| DDIM-40 steps | 9.14 | $1.24\times$ | 2.39 | 4.28 | 0.80 | 0.59 |
| DDIM-25 steps | 5.73 | $1.96\times$ | 3.01 | 4.60 | 0.79 | 0.58 |
| DDIM-20 steps | 4.58 | $2.42\times$ | 3.48 | 4.64 | 0.79 | 0.56 |
| FORA | 4.13 | $2.12\times$ | 3.88 | 6.74 | 0.79 | 0.56 |
| ToCa | 4.97 | $2.09\times$ | 3.05 | 4.70 | 0.79 | 0.57 |
| DuCa | 4.94 | $2.10\times$ | 3.04 | 4.70 | 0.79 | 0.57 |
| **DiffSparse** | 4.97 | $2.07\times$ | **2.81** | **4.61** | **0.80** | **0.59** |

**Results on Class-Conditional Image Generation.** Table 2 compares faster sampler DDIM with fewer steps, FORA, ToCa, DuCa and DiffSparse. Our method achieves a better speed–accuracy balance by reallocating computation to the most important layers. At the same acceleration ratio, DiffSparse improves the FID from 3.05 to 2.81, outperforming ToCa by 8% at $2.07\times$ acceleration. demonstrating its ability to preserve detail and improve image fidelity in diffusion model acceleration.

Table 3: Comparison in text-to-video generation for Wan2.1-1.3B with 20 sampling steps on VBench.

| Method | MACs (T) ↓ | Speedup ↑ | VBench ↑ |
|---|---|---|---|
| Wan 2.1 - 1.3B | 43.866 | $1.00\times$ | 43.82 |
| 50% steps | 21.933 | $1.86\times$ | 43.14 |
| DuCa (R = 54%) | 20.332 | $1.69\times$ | 43.56 |
| DuCa (R = 59%) | 18.124 | $1.68\times$ | 43.30 |
| **DiffSparse** | 18.124 | **$2.05\times$** | **43.83** |

Table 4: Comparison on PixArt-$\alpha$ using 20 sampling steps at $512\times512$ resolution.

| Method | MACs (T) ↓ | FID ↓ | CLIP ↑ |
|---|---|---|---|
| PixArt-$\alpha$ | 10.851 | 21.95 | 0.164 |
| 50% steps | 5.426 | 25.05 | 0.163 |
| ToCa | 5.993 | 23.02 | 0.165 |
| **DiffSparse** | 5.986 | **22.42** | **0.165** |

**Results on Text-to-Video Generation.** Table 3 presents a comparison between DiffSparse and DuCa (Zou et al., 2024) on Wan2.1-1.3B (Wan et al., 2025) using 20 sampling steps. The methods are comprehensively evaluated across 16 aspects defined in VBench (Huang et al., 2024). We adopt

DuCa's norm-based token ranking compatible with FlashAttention (Dao et al., 2022) for faster inference. DiffSparse achieves the highest VBench score while minimizing computational cost and inference time. At the same compression ratio, it delivers greater speedup by skipping partial layers with zero sparsity, and its adaptive, layer-wise sparsity allocation preserves model quality.

## 4.3 ABLATION STUDIES

**Comparison of Two Stage Training.** In this work, we adopt a two-stage training strategy. The first stage independently trains cost matrices for full-step and layer sparsity. In the second stage, the learned full-step cost is merged into the layer sparsity optimization, and the layer sparsity is subsequently fine-tuned. This design enables the model to initially leverage the full-step to correct errors and to learn layer sparsity cost values, followed by a gradual reduction of the full-step influence. Results proved that two-stage approach achieves better performance, with an FID of 26.91 compared to 27.40 from the single-stage baseline.

**Comparison of Important Scores.** Table 5 compares three importance scores: attention (Equation 10), cosine similarity and the $\ell_2$ norm. The cosine similarity is computed between the current input token and cached tokens. The $\ell_2$ norm is the norm value of input tokens. The attention-based score attains the best FID, followed by the similarity measure, which captures token redundancy effectively. Norm-based scoring introduces noise and performs worst, confirming that accurate importance estimation is critical for optimal token selection.

Table 5: Ablation study on token importance metrics.

| Method | Base. | w/ DiffSparse |
|---|---|---|
| Norm | 29.05 | 28.89 (-0.16) |
| Similarity | 29.00 | 28.07 (-0.93) |
| Attention | 28.35 | **26.91 (-1.44)** |

Table 6: Ablation study on distillation loss functions.

| Method | FID ↓ | CLIP ↑ |
|---|---|---|
| L2 | 27.68 | 0.164 |
| SSIM | 27.46 | 0.164 |
| LPIPS | **26.91** | **0.164** |

Table 7: Ablation study of sparse interval.

| Interval | $|S|$ | FID ↓ | CLIP ↑ |
|---|---|---|---|
| 0.1 | 11 | 27.96 | 0.163 |
| 0.125 | 9 | 27.91 | 0.163 |
| 0.25 | 5 | **26.91** | 0.164 |
| 0.5 | 3 | 27.54 | **0.164** |
| 1.0 | 2 | 28.22 | 0.162 |

Table 8: Ablation of warm-start strength $\delta$.

| $\delta$ | FID ↓ | CLIP ↑ |
|---|---|---|
| 0 | 27.40 | 0.163 |
| 5 | 27.01 | 0.164 |
| 10 | **26.91** | **0.164** |
| 20 | 26.95 | 0.164 |

**Comparison of Training Losses.** We compare L2, SSIM (Wang et al., 2004), and LPIPS losses in Table 6. LPIPS outperforms the others, yielding the best FID. L2 loss penalizes pixel-wise squared errors and often produces overly smooth images that lack fine details. SSIM enforces local structural similarity but may over-penalize perceptually good images that differ spatially from the original. By measuring distances in a learned perceptual feature space, LPIPS avoids these pitfalls and better preserves image quality during training.

**Comparison of Sparse Intervals.** We distribute sparsity uniformly across layers by token count and evaluate different granularity settings in Table 7. A granularity of 0.125 yields minimal within-layer variation, which hinders convergence, while 0.5 limits the range of sparsity choices. The optimal granularity is 0.25, producing sparsity rates [0, 0.25, 0.50, 0.75, 1.0] (corresponding to candidate token counts of [0, 64, 128, 192, 256] for a sequence length of 256) and delivering the best performance.

**Generalization on Higher Resolution Models.** As the token sequence length increases with image resolution, peak memory usage during training grows substantially, even though the size and computational cost of our cost matrix remain unchanged. This makes direct training at very high resolutions impractical. To address this, we investigate whether a sparsity predictor trained at lower resolution can be **transferred to higher resolution without retraining**. As shown in Table 4, the sparsity predictor learned at 256 × 256 resolution achieves a lower FID than ToCa on 512 × 512 images while maintaining a comparable CLIP-Score to the original PixArt model. These results demonstrate that our method generalizes effectively to higher resolutions, enabling model acceleration with limited memory and training cost.

**Compared with GA Search.** We compared DiffSparse against traditional search methods such as random search and genetic algorithms and found that in the vast sparsity space they underperform. After 1,000 iterations on 500 images, these methods yield FID scores of 28.34 and 27.94, respectively, compared with 26.91 for DiffSparse. Moreover, they require about 16 hours, whereas DiffSparse completes training in roughly 4 hours. These results show that our differentiable learning framework discovers more effective layer-wise sparsity allocations and delivers superior acceleration.

**Comparison of Warm-Start Constant $\delta$.** Algorithm 1 uses a warm-start constant $\delta = 10$ for the two-stage optimization. Intuitively, $\delta$ injects the Stage-1 prior (the timesteps selected to remain full-step) into Stage 2 by lowering the cost of the "full" candidate at those timesteps. In effect, a larger $\delta$ more strongly encourages preserving full computation at the Stage-1 selected steps. To quantify this effect we evaluated $\delta \in \{0, 5, 10, 20\}$. Table 8 reports the results on PixArt-$\alpha$ with $T = 20$. A moderate warm-start ($\delta = 10$) recovers most of the benefit, while $\delta = 0$ (no warm-start) removes the Stage-1 prior and yields noticeably worse performance.

## 4.4 QUALITATIVE ANALYSIS

**Visualization of Generated Images.** We provide detailed visual comparisons among our proposed method, ToCa, and the original PixArt-$\alpha$ across various sparsity ratios. in Figure 2 reveals that DiffSparse consistently maintains high fidelity, even under aggressive pruning conditions. Moreover, our DiffSparse effectively preserves the semantic content of the text prompt, ensuring that the generated images remain closely aligned with the original descriptions. In contrast, baseline methods exhibit noticeable degradation in both visual quality and text-image alignment at higher pruning ratios, further highlighting the strength and efficiency of DiffSparse.

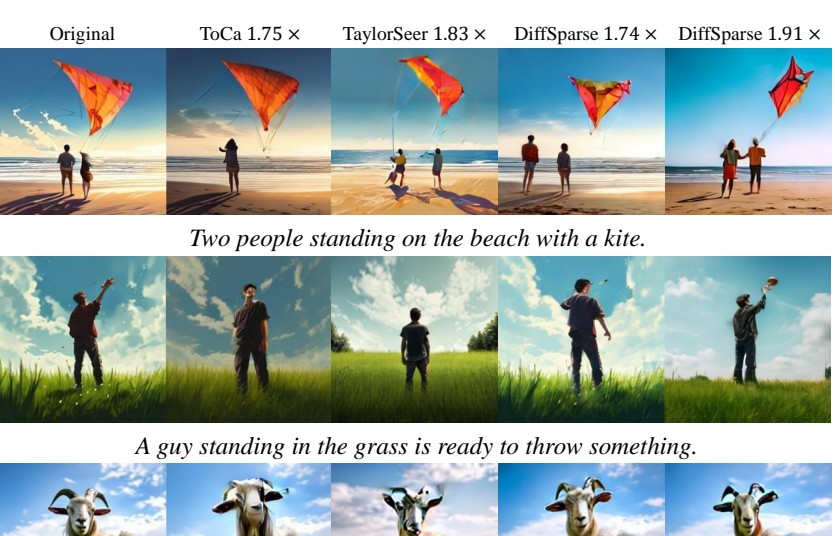

| Original | ToCa 1.75 × | TaylorSeer 1.83 × | DiffSparse 1.74 × | DiffSparse 1.91 × |

*Two people standing on the beach with a kite.*

*A guy standing in the grass is ready to throw something.*

*A goat with horns is standing in a grassy field.*

Figure 2: Comparison of our method with the baseline (PixArt-$\alpha$ with DPM-Solver++ using 20 steps) and existing methods under different acceleration rates.

## 5 CONCLUSION

We introduce a learnable token sparsity allocation framework to accelerate diffusion transformers. By formulating sparsity allocation as a dynamic programming problem and employing a two stage training strategy, our method substantially reduces computational cost while preserving generative quality. Extensive experiments across various foundation models and datasets demonstrate improved acceleration ratios without compromising image quality of our method.

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

# A APPENDIX

## A.1 ETHICAL STATEMENT

Generative models have shown impressive capabilities in content creation (Chen et al., 2023; Rombach et al., 2022), but their high inference costs hinder rapid deployment. Our method offers an efficient acceleration strategy for diffusion models, achieving near-lossless speedup without retraining and maintaining compatibility with various architectures. This generalizability makes it well-suited for fast deployment on mobile and edge devices.

However, generative models pretrained on large-scale internet data may reflect inherent social biases and stereotypes. There is also potential for misuse, such as in DeepFake (Lyu, 2020) creation, which can cause serious societal harm. As the cost of generation decreases, the risk of irresponsible use increases. Therefore, it's essential to establish regulations, foster a well-governed community, and provide clear usage guidelines to ensure the responsible application of generative technologies.

## A.2 REPRODUCIBILITY STATEMENT

To support reproducibility, we provide detailed pseudocode for the proposed method (Appendix A.5.3), full training and evaluation protocols including all hyperparameters and optimizer settings (Section 4.1 and Appendix A.5), dataset descriptions and preprocessing steps, and the computing environment for experiments (Section 4.1). Where applicable, we report evaluation metrics and include instructions sufficient to reproduce the experimental pipelines described in the main text and appendices. We plan to release anonymized code, trained model checkpoints, and exact run scripts upon paper acceptance to facilitate full replication.

## A.3 THE USE OF LARGE LANGUAGE MODELS (LLMS)

We did not rely on LLMs for research ideation, experiment design, data analysis, or the generation of technical content. Any use of LLMs was limited to minor, general-purpose editorial assistance (proofreading, grammar, phrasing, and formatting suggestions); all such suggestions were reviewed and revised by the authors. The paper's conceptual contributions, algorithms, experiments, results, and conclusions were produced solely by the authors, and no LLM is credited as a contributor.

## A.4 MORE DISCUSSION WITH EXISTING WORKS

### A.4.1 MORE RELATED WORKS

**Diffusion Transformer Models.** Recent work has improved the efficiency and scalability of transformer-based diffusion models. Hybrid CNN–transformer architectures (Saharia et al., 2022) combine local inductive biases with global attention, and transformer-based video generation (Ho et al., 2022) demonstrates strong temporal modeling. These results establish transformers as a versatile backbone for diffusion, motivating efforts on optimization, faster inference, and stronger conditional generation. Nevertheless, the iterative denoising loop still incurs substantial computational overhead that limits industrial deployment.

**Acceleration of Diffusion Models.** Several recent methods target inference cost directly: EOC leverages prior knowledge to improve caching (Qiu et al., 2025), while designs such as UniCP and RAS further boost efficiency (Sun et al., 2025; Liu et al., 2025d). DyDiT (Zhao et al., 2024) accelerates inference by skipping unimportant tokens and slimming per-layer width, whereas DiffSparse reduces compute by *reusing* cached features. The two strategies are complementary and can be combined for larger speedups. DiffSparse computes token importance with a training-free compositional-attention score and learns a compact layer-wise predictor, leading to much faster convergence (on the order of $10^3$ iterations versus DyDiT's $\sim 2 \times 10^5$ fine-tuning steps). Moreover, by optimizing a global $T$-step objective with a dynamic-programming solver, DiffSparse coordinates sparsity across timesteps and layers and is validated across multiple architectures and generation tasks.

**Comparison with Search-based and Training-based Methods.** In our main configurations, the differentiable optimization requires $\approx 4$ hours of training versus $\approx 16$ hours for a genetic-algorithm

search baseline, DiffSparse attains better FID while using less optimization time. The learned sparsity predictor is compact (size $(T \times L) \times |S|$) and often transfers from $256 \times 256$ training to $512 \times 512$ evaluation, reducing the need for retraining at higher resolutions. By contrast, distillation-based pipelines can demand orders of magnitude more compute: reported distillation efforts (DMD2) involve $\mathcal{O}(10^3 - 10^4)$ GPU·hours (e.g., SD1.5: 1,664 GPU·hr; SDXL: 8,192 GPU·hr), far exceeding the cost of our method and frequently relying on private data. Importantly, DiffSparse also improves some distilled models: for example, on a 4-step distilled model (FLUX.1-schnell) we observe a $1.81\times$ speedup with no measurable quality drop (see Table 9).

## A.5 MORE IMPLEMENTATION DETAILS

### A.5.1 TOKEN SELECTOR

We rank tokens using a composite importance score that integrates four criteria: self-attention influence, cross-attention influence, cache reuse frequency, and uniform spatial distribution. This composite score, which has demonstrated effective in prior work (Zou et al., 2025), is defined for each token $\hat{x}_i$ as follows:

$$S(\hat{x}_i) = \mathcal{B}\Big(\lambda_1 \, s_1(\hat{x}_i) + \lambda_2 \, s_2(\hat{x}_i) + \lambda_3 \, s_3(\hat{x}_i)\Big), \tag{10}$$

where $s_1(\hat{x}_i) = \sum_{j=1}^{N} \alpha_{ij}$ quantifies the self-attention contribution of token $\hat{x}_i$, with $\alpha_{ij}$ being the $(i, j)$-th element of the normalized self-attention matrix. A higher value indicates that the token exerts significant influence on others, meaning error in its representation may easily propagate. The term $s_2(\hat{x}_i) = -\sum_{j=1}^{N} o_{ij} \log(o_{ij})$ represents the entropy of the cross-attention weights $o_{ij}$, measuring how the control signal influences token $\hat{x}_i$, with lower entropy indicating more focused guidance. Additionally, $s_3(\hat{x}_i) = n_i$ denotes the number of times token $\hat{x}_i$ has been reused from the cache since its last computation, where a higher $n_i$ suggests possible accumulated errors, thus necessitating a fresh computation. The spatial bonus function $\mathcal{B}(\cdot)$ promotes a uniform spatial distribution of the selected tokens by adding a bonus value $\lambda_4$ to the score of $\hat{x}_i$ if it has the highest composite score within its $k \times k$ neighborhood. For each layer, tokens are ranked in descending order based on $S(\hat{x}_i)$, and the top $K$ tokens are selected for computation and cache updates according to a predefined sparsity ratio $R$.

We adopt the hyperparameter settings recommended by ToCa (Zou et al., 2025) and DuCa Zou et al. (2024) (which were shown to be optimal for that setup) and therefore do not include ablation experiments for these parameters, since tuning them is not central to our contribution. Specifically, for PixArt-$\alpha$, we set $\lambda_1 = 0.0$, $\lambda_2 = 1.0$, $\lambda_3 = 0.25/3$, $\lambda_4 = 0.4$, and $k = 4$. For DiT, we use $\lambda_1 = 1.0$, $\lambda_2 = 0.0$, $\lambda_3 = 0.25/3$, $\lambda_4 = 0.6$, and $k = 2$. For FLUX.1-schnell, we set $\lambda_1 = 0.0$, $\lambda_2 = 1.0$, $\lambda_3 = 0.25/3$, $\lambda_4 = 0.4$, and $k = 4$.

Besides, for Wan2.1, we select tokens with smaller norms in their value matrix as substitutes for those with high attention map scores, following a strategy shown to be effective in DuCa Zou et al. (2024). Notably, our method does not introduce a new token-selector. Instead, it can be applied to existing token-selection methods and uses a differentiable sparsity-cost matrix to assign the model an optimal sparsity level. As shown in Table 5, across various token-importance metrics our approach consistently yields substantial gains.

### A.5.2 LAYER SPARSITY COST PREDICTOR

We define a sparsity router for each layer. For PixArt-$\alpha$ and Wan2.1 model, each transformer block consists of a self-attention layer, a cross-attention layer, and an MLP layer, with each layer being assigned an individual sparsity value. In contrast, the DiT model does not include a cross-attention layer, thus the corresponding predictor for cross-attention layer is removed. Additionally, the FLUX model contains an image MLP layer, a text MLP layer, and a standard MLP layer, each of which is assigned its own sparsity value.

### A.5.3 TWO-STAGE TRAINING

We present the pseudocode for our two-stage training algorithm in Algorithm 1, illustrating the training details of our sparsity cost predictor.

---

**Algorithm 1** Two-Stage Training Strategy for Cost Matrix Optimization

---

**Input:** Step cost matrix $C_f \in \mathbb{R}^{T \times 2}$; Layer sparsity cost matrix $C_l \in \mathbb{R}^{(L \times T) \times |S|}$; Total steps $T$; Number of layers $L$; Candidate set $S$; Desired full-step count $|T_f|$; Mutation constant $\delta = 10$.
**Stage 1: Initialization and Warm-Starting**
Solve $C_f$ via dynamic programming to obtain optimal full-step set $T_f$.
Integrate $C_f$ into the tuned $C_l$ to form the unified cost matrix:
**for** each $t \in T_f$ **do**
   **for** $l = 1$ to $L$ **do**
      Update cost: $C_l^{(t,l,N)} \leftarrow C_l^{(t,l,N)} - \delta$.
   **end for**
**end for**
**Stage 2: Unified Cost Optimization**
Fine-tune the integrated $C_l$ using differentiable cost interactions to systematically redistribute FLOPs across sampling steps.
**Output:** Optimized layer cost matrix $C_l$.

---

### A.5.4 SEARCH-BASED APPROACHES

In this paper, we compare our method with search-based approaches. For the GA algorithm, we start by initializing a population of 50 $(T * L)$ layer-sparsity vectors that satisfy the sparsity requirements. Each candidate is evaluated using its FID value, which serves as the fitness score. In subsequent iterations, we select the best-performing individuals as parents for crossover operations and introduce mutations with a probability of 0.01 to maintain population diversity. This iterative process continues until the individual with the highest fitness score is identified. Besides, the random search algorithm generates a population of candidates that meet the sparsity requirements in a completely random manner. Their fitness is also evaluated using the FID value, and the optimal solution is updated iteratively until the maximum number of iterations is reached.

### A.5.5 ABOUT THE RETRAINING REQUIREMENT

If you change the *model architecture* significantly (e.g., different number of layers $L$ or a different block structure), retraining or at least fine-tuning is required when the *temporal* or *architectural* axes change ($T$ or $L$) because its parameters are tied to $(T, L, |S|)$, but not usually when only token length (image resolution) increases. Given the modest one-time training cost (4–10 GPU-hours in our experiments) and the measurable quality, speed improvements, we believe the overhead is justified for deployed models where inference cost matters.

## A.6 MORE EXPERIMENTS

### A.6.1 COMPARISON ON DISTILLED MODEL

Feature caching leverages redundancy across timesteps but provides little benefit for distilled diffusion models with only one or two steps. Nevertheless, we still evaluate DiffSparse on FLUX.1-schnell (Labs, 2024) with 4 steps at 256×256 resolution on the PartiPrompts (Yu et al., 2022) dataset. As Table 9 shows, DiffSparse attains the same acceleration rate as ToCa but yields a higher Image Reward, confirming its effectiveness in reallocating computation to the most critical layers and delivering lossless speedup.

Table 9: Comparison in text-to-image generation for FLUX.1-schnell on PartiPrompts.

| Method | MACs (T)↓ | Image Reward↑ |
|---|---|---|
| FLUX.1-schnell | 13.247 | 1.064 |
| 75% Steps | 9.936 | 1.063 |
| ToCa | 7.313 | 1.063 |
| **DiffSparse** | 7.316 | **1.184** |

### A.6.2 ABLATIONS OF THE ATTENTION-BASED SCORE

Table 5 compares three importance metrics (attention-based score, cosine similarity, $\ell_2$ norm) and shows the attention-based score performs best overall. In Table 10, we further remove one component at a time ($-s_1, -s_2, -s_3, -B$) to answer how much each term contributes independently.

Table 10: Ablations of the attention-based score in text-to-image generation for PixArt-$\alpha$.

| Variant | FID↓ | CLIP↑ |
|---|---|---|
| DiffSparse | **26.91** | **0.164** |
| $-s_1$ (self-attention influence) | 27.11 | 0.164 |
| $-s_2$ (cross-attention focus) | 27.48 | 0.163 |
| $-s_3$ (cache-reuse frequency) | 27.23 | 0.164 |
| $-B$ (spatial bonus) | 27.05 | 0.164 |

## A.7 QUALITATIVE ANALYSIS

### A.7.1 VISUALIZATION OF LAYER SPARSITY

Figure 3 shows the learned layer wise sparsity allocation for PixArt-$\alpha$ at 256×256 resolution with 20 sampling steps under 1.74× speedup, with the first step omitted because it is always fully computed in cache-based acceleration methods. In the self attention layers, sparsity is higher (that is, the layers are more cacheable) in early time steps and shallow layers, while in the cross attention layers sparsity is lower in later time steps and deeper layers, suggesting that textual semantics are most important in the initial layers. The MLP layers receive more computation in early steps and shallow layers, with reduced sparsity in deep layers at early steps and in shallow layers at later steps. In addition, Figure 3 demonstrates that our method can redistribute the computation across all steps, reducing the dependence on fully computed steps. It shows that additional resources are allocated to MLP layer, this might be attributed to its ability of correct errors introduced by caching.

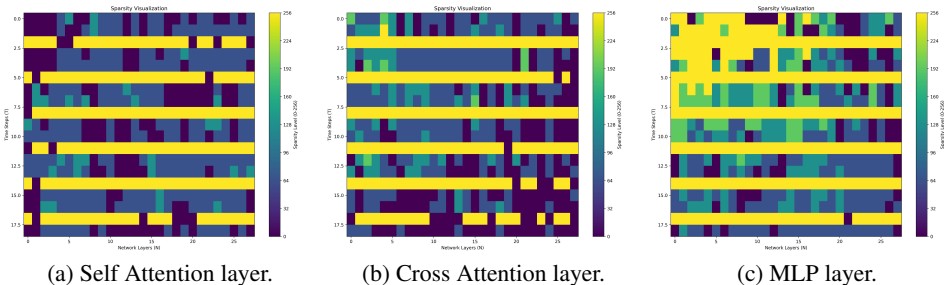

(a) Self Attention layer.  (b) Cross Attention layer.  (c) MLP layer.

Figure 3: Visualization of predicted layer sparsity of PixArt-$\alpha$ with 20 steps. In the figure, the x-axis denotes different network layers, the y-axis denotes sampling time steps, and the color gradient from blue to yellow indicates increasing sparsity.

### A.7.2 MORE VISUALIZATION OF GENERATED IMAGES

Figures 4 and 5 present further visualizations, including additional comparison samples and higher-resolution results. They confirm that our approach delivers markedly higher acceleration ratios compared to the baseline, while preserving performance quality.

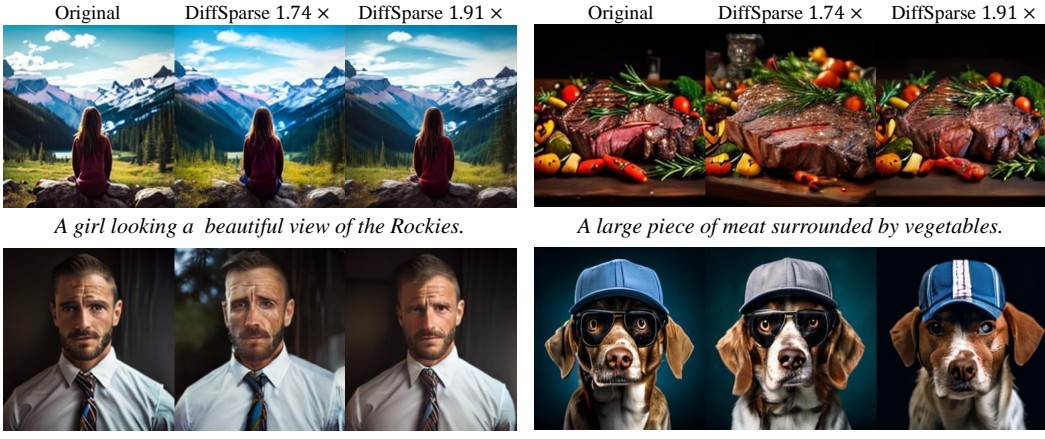

Figure 4: Comparison of our method with the baseline (PixArt-$\alpha$ with DPM-Solver++ using 20 steps) under different acceleration rates.

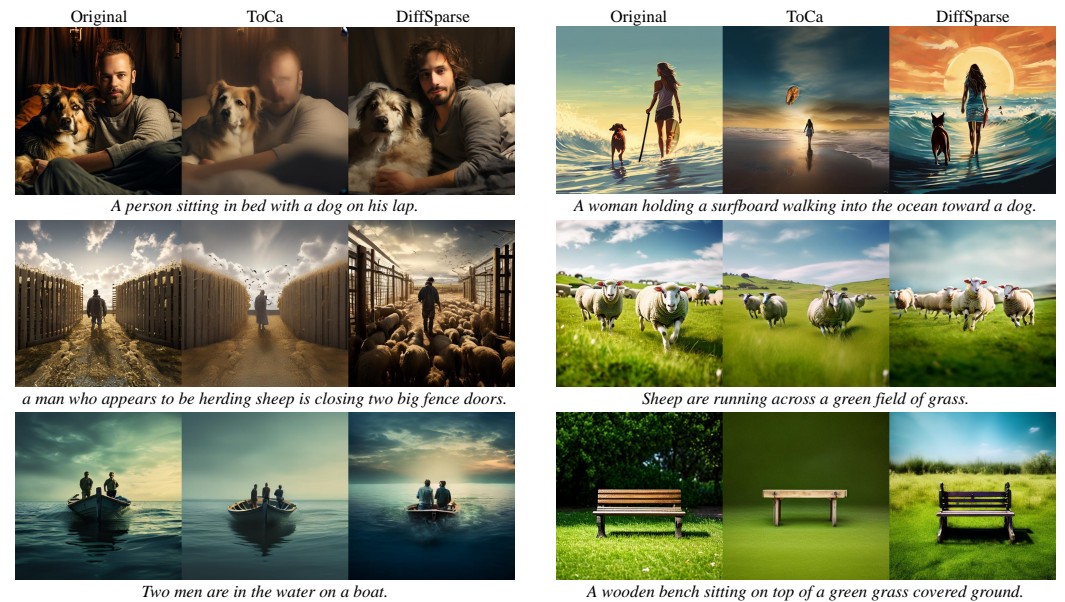

Figure 5: Comparison between our DiffSparse, and ToCa with the baseline (PixArt-$\alpha$ with DPM-Solver++ using 20 steps under 512×512 resolution).

