# OpenReview forum: "DiffSparse: Accelerating Diffusion Transformers with Learned Token Sparsity"
_ICLR.cc/2026/Conference — ICLR 2026 Poster_

### Official Review · Reviewer_wjJ7 · 2025-10-30

**Soundness:** 3
**Presentation:** 1
**Contribution:** 3
**Rating:** 4
**Confidence:** 3

**Summary:**

This paper focuses on the issue of high computational cost of diffusion models, and proposes the **DiffSparse** framework, which achieves acceleration through optimization via learnable layer-wise sparsity. This method eliminates the need for manual adjustment of sparsity parameters and realizes automated sparsity allocation through end-to-end learning, making it more efficient than traditional feature caching methods that rely on manual sparsity allocation. DiffSparse performs excellently on a variety of diffusion models and possesses cross-resolution generalization capability. Meanwhile, compared with search methods such as genetic algorithms, it also features higher training efficiency.

**Strengths:**

### 1. Innovative Method Design & Problem Formulation
Redefines diffusion transformer acceleration as a **differentiable layer-wise sparsity allocation problem**, abandoning the manual sparsity schedules of prior token cache methods (e.g., ToCa). It combines a learnable sparsity cost predictor, dynamic programming solver, and adaptive token selector into a unified framework. Additionally, it eliminates predefined full-step computation via a two-stage training strategy to unlock token caching’s full acceleration potential, and achieves cross-resolution generalization where low-resolution-trained sparsity predictors work for high-resolution models without retraining.

### 2.  Robust Experimental Validation
It conducts comprehensive experiments, validating on 4 models (DiT-XL/2, PixArt-α, FLUX, Wan2.1) across image/video generation—for example, PixArt-α achieves 54% cost reduction (1.91× speedup) with better FID than ToCa. Thorough ablation studies confirm the optimality of two-stage training, LPIPS loss, and 0.25 sparsity interval. It also outperforms search methods like genetic algorithms with 4× faster training.

**Weaknesses:**

### 1. Insufficient Analysis of Computational Overhead
- **Unreported training/inference overhead of DiffSparse components**: The paper claims the dynamic programming solver has low complexity $O((L \cdot T)^2 \cdot |S|)$ but does not report the **actual runtime overhead** of the sparsity cost predictor or the dynamic programming solver during inference.
- **Lack of memory consumption breakdown**: While noting cross-resolution generalization reduces memory usage, the paper does not provide quantitative data (e.g., peak memory for 256×256 vs. 512×512 training with DiffSparse) or compare it to baselines.

### 2. Limited Generalization Validation
- **No testing on ultra-low-step models**: The paper excludes distillation methods but does not test DiffSparse on **1–2 step distilled diffusion models** (e.g., SDXL-Turbo).

### 3. Incomplete Ablation of Key Components
- **No ablation of dynamic programming solver necessity**: The paper claims the dynamic programming solver optimizes global sparsity, but it does not compare against a "no solver" baseline (e.g., using the cost predictor’s output directly without global constraint optimization). This fails to prove the solver’s contribution to performance.

**Questions:**

### **1. Similarity of Dynamic Sparsity Allocation in Other Feature Caching-Based Diffusion Acceleration Works**
 Are there other papers that use Feature caching to accelerate diffusion models and have similar dynamic sparsity allocation ideas?

### **2. Feature Recomputation and Utilization Logic After Sparsity Determination**
Although the method in this paper aims to achieve dynamic sparsity allocation, it lacks mention of how features are recomputed and utilized after sparsity is determined. For example, does this paper directly adopt previous work, or it make modifications based on previous methods?

### **3. Input, Cost Definition, and Calculation Logic of Learnable Sparsity Cost Predictor**
What is the input of the Learnable Sparsity Cost Predictor? What is "Cost" defined as, and how is the Cost of a specific layer of the model at a certain timestep calculated?

### **4. Rationale for Using Formula (9) to Integrate Step and Layer Costs**
 Formula (9) is intended for warm-start. Why can it be used to integrate step and layer costs? I cannot see the logic connection for now.

### **5. Lack of Performance Comparison with Other Non-Feature-Caching-based Acceleration Methods**
Although this paper focuses on accelerating diffusion models using Feature Caching, the experiments lack performance comparisons with non-Feature caching acceleration methods, such as Model pruning, Rectified Flow, and knowledge distillation.

---

> ### Author Response · Authors · 2025-11-23
> **Response to Reviewer wjJ7**
>
> Thank you for the constructive comments. We appreciate your recognition of (1) the novel formulation of layer-wise differentiable sparsity allocation, and (2) the broad experimental validation across four transformer-based models and image/video settings. Below we address the concerns.
>
> ---
>
> ## Weakness 1 — Insufficient analysis of computational overhead
>
> - The sparsity cost predictor is a compact learnable matrix with size $(T × L) × |S|$ and outputs a normalized cost tensor $C ∈ ℝ^{(T×L)×|S|}$. Its parameter count and memory overhead are independent of the token sequence length $N$.
>
> - The dynamic programming (DP) solver used to enforce the global sparsity constraint has worst-case time complexity $O((L·T)^2 · |S|)$. For PixArt-α, with $|S|=5, T=20, L=28$, the end-to-end training (including DP optimization and fine-tuning) completes in ≈**4 hours** . The DP stage itself runs in roughly **30 s** for these configs. Importantly, DP is an **offline** training-time step and is **not** run at inference; inference uses the precomputed binary masks derived by DP.
>
> - We measured peak GPU memory during training for PixArt-α: at $256×256$ the peak is **61.82 GB**, and at $512×512$ it is **92.46 GB**. The $512×512$ results in the paper were obtained by **transferring** the predictor trained at $256×256$ (no retraining at $512×512$), which leverages the predictor’s independence from $N$ and reduces practical memory burden. Additional memory-saving techniques (mixed precision, gradient checkpointing, FSDP) can further reduce peak memory.
>
> ---
>
> ## Weakness 2 — Limited Generalization Validation (ultra-low-step / distilled models)
>
> Feature-caching generally yields little benefit for heavily distilled ultra-low-step samplers (1–2 steps) because those samplers remove most temporal redundancy. Nevertheless, we evaluated DiffSparse on a 4-step distilled model (**FLUX.1-schnell**); results are in Table 9. On this model DiffSparse attains a **1.81×** speedup with no measurable quality degradation.
>
> ---
>
> ## Weakness 3 — Incomplete ablation of the dynamic programming solver
>
> We use a DP solver to convert the learned cost matrix into a globally feasible per-layer sparsity allocation under an overall compute budget. DP reasons about inter-layer and inter-step trade-offs so the final sparsity map is globally optimal w.r.t. the learned costs, rather than being the result of independent per-layer decisions.
>
> To illustrate the benefit: a hand-designed allocation in the spirit of ToCa (different fixed sparsity for shallow vs deep layers) yields **FID = 28.35** on PixArt-α. DiffSparse (DP-derived allocation) achieves **FID = 26.91**, i.e., a gap of **1.44 FID**, showing that global optimization materially improves sample quality over heuristic allocations.
>
> ---
>
> ## Questions
>
> 1. **Q1 — Similarity to other works:**
>    There is a growing body of feature-caching and dynamic-caching work, but none use the same dynamic, global sparsity-allocation strategy that jointly optimizes across layers and timesteps as DiffSparse does.
>
> 2. **Q2 — Feature recomputation & utilization logic:**
>    DiffSparse follows standard token-cache update rules used in prior token-level caching works: tokens selected for reuse bypass recomputation while others are recomputed and the cache is updated for recomputed tokens. The novelty is learning *when* and *how many* tokens each layer should recompute across timesteps.
>
> 3. **Q3 — Input and definition of “Cost”:**
>    The predictor is a learnable table of size $(T×L) × |S|$ (a scalar cost for every $(t,l,s)$), it does **not** take per-image token features as input. Each entry $C(t,l,s)$ estimates the expected perceptual penalty of applying sparsity configuration $s$ at layer $l$ and timestep $t$.
>
> 4. **Q4 — Rationale for Formula (9):**
>    Stage-1 solves the step cost matrix $C_f$ via DP to identify $T_f$ (timesteps to keep full). To bias the layer predictor toward preserving full computation at these timesteps, we subtract a margin $δ$ from the corresponding layer-cost entries (Eq. (9)), lowering the cost of the “full” candidate there. This warm-start preserves inter-layer cost ranking while encouraging full steps in the subsequent joint optimization. We will clarify this intuition in the revision.
>
> 5. **Q5 — Comparison with non-feature-caching acceleration:**
>    Model pruning and distillation typically require large-scale extra training; their compute cost is often orders of magnitude larger than our optimization cost, and many distillation pipelines rely on private training data. Rectified Flow is an orthogonal advance in sampler design (improves sampling dynamics rather than per-step compute or feature reuse). We evaluate DiffSparse on distilled / Rectified Flow-based samplers (e.g., FLUX.1-schnell); Table 9 shows DiffSparse attains **1.81×** speedup with no quality loss, demonstrating complementarity with sampler-level improvements.

---

> > ### Author Response · Authors · 2025-11-28
> > **Kindly Request for Your Feedback!**
> >
> > Dear Reviewer wjJ7,
> >
> > We appreciate your careful reading and constructive comments on our submission. Below we summarize the key clarifications and new data from our rebuttal that directly address your concerns, and respectfully ask for your feedback on whether these responses alleviate your main reservations.
> >
> > **Summary of our responses**
> >
> > 1. **Computational overhead & practicality**
> >
> >    * The sparsity cost predictor is a compact learnable table of size $(T \times L) \times |S|$ and is independent of token sequence length $N$ (so its parameters and memory do **not** scale with $N$).
> >    * The DP solver used to enforce the global budget has worst-case time complexity $O((L\cdot T)^2 \cdot |S|)$. For our PixArt-α configs ($|S|=5, T=20, L=28$) end-to-end training (including DP + fine-tuning) completes in ≈ **4 hours**, while the DP stage alone runs in roughly **30 s**. Crucially, DP is an **offline, training-time** step — inference uses precomputed binary masks and does **not** run DP.
> >    * Measured peak GPU memory for PixArt-α: **61.82 GB** at $256\times256$ and **92.46 GB** at $512\times512$. To obtain the $512\times512$ results we **transferred** the predictor trained at $256\times256$ (no retraining at $512\times512$), leveraging predictor independence from $N$. Standard memory-saving techniques (mixed precision, checkpointing, FSDP) can further reduce peak usage.
> >
> > 2. **Generalization to distilled models**
> >
> >    * Feature-caching provides little benefit for extremely distilled 1–2-step samplers because temporal redundancy is largely removed. Nonetheless, we evaluated DiffSparse on a 4-step distilled sampler (**FLUX.1-schnell**): DiffSparse achieves a **1.81×** speedup with no measurable quality degradation (Table 9), showing complementarity with distillation models.
> >
> > 3. **Role and ablation of the DP solver**
> >
> >    * DP converts the learned per-(t,l,s) cost matrix into a globally feasible per-layer sparsity allocation under the overall compute budget, reasoning jointly across timesteps and layers (rather than making independent per-layer choices).
> >    * A hand-designed, ToCa-style allocation yields **FID = 28.35** on PixArt-α, while the DP-derived DiffSparse allocation yields **FID = 26.91** — an improvement of **1.44 FID**, demonstrating that global optimization materially improves sample quality.
> >
> > 4. **Other clarifications**
> >
> >    * **Cost predictor input:** the predictor is a learnable matrix (it does not take per-image token features as input). Each entry $C(t,l,s)$ estimates expected perceptual penalty of applying sparsity candidate $s$ at $(t,l)$.
> >    * **Formula (9) rationale:** Stage-1 finds timesteps to keep full via DP, we subtract a margin $δ$ from corresponding layer-cost entries to bias the joint optimizer toward preserving full steps there — a warm-start that preserves inter-layer cost ranking while encouraging full steps.
> >    * **Comparison to pruning/distillation:** those approaches usually require large-scale extra training (often with private data) and far larger compute cost. DiffSparse’s optimization is lightweight and complementary to sampler advances (e.g., Rectified Flow, distillation models).
> >
> > We hope these points address the major concerns raised and clarify the design choices and empirical trade-offs.
> >
> > If our explanations and the additional results resolve your concerns, we would be grateful if you could reconsider your rating. If any part of our rebuttal remains unclear or you would like additional ablations, we are happy to provide them.
> >
> > Thank you very much for your time and thoughtful review. We would greatly appreciate your feedback on the rebuttal and whether the above responses sufficiently address your concerns.
> >
> > Sincerely,
> > The Authors

---

> ### Comment · Area_Chair_PaqH · 2025-11-28
>
> Dear Reviewer, the discussion period is about to close. We kindly ask you to participate in the discussion or update your score based on the authors' rebuttal before the deadline. Thank you for your time and valuable contribution!

---

### Official Review · Reviewer_c1BY · 2025-10-30

**Soundness:** 3
**Presentation:** 3
**Contribution:** 3
**Rating:** 6
**Confidence:** 5

**Summary:**

This paper introduces DiffSparse, a framework to accelerate diffusion transformer models by optimizing layer-wise token sparsity during the sampling process. It employs a learnable sparsity cost predictor and dynamic programming to allocate computation efficiently, eliminating the need for manual sparsity tuning and full-step computations. The proposed two-stage training strategy enhances acceleration while preserving image generation quality. Extensive experiments on various models demonstrate significant speedups (up to 1.91×) with improved or comparable performance to state-of-the-art methods.

**Strengths:**

+ The proposed differentiable layer-wis sparsity optimization is novel and effective.

**Weaknesses:**

1. What is the training cost for FLUX and Wan?

2. The paper would benefit from reorganization. For instance, the related work section is overly long and could be trimmed and it would be better to include visualizations directly in the main paper to enhance clarity and accessibility.

3. The discussion should also include other token-based acceleration methods, such as DyDiT [1].

[1] Dynamic Diffusion Transformer, 2025 ICLR

**Questions:**

see weakness

---

> ### Author Response · Authors · 2025-11-23
> **Response to Reviewer c1BY**
>
> Thank you for the positive feedback and constructive suggestions. We address your concerns and questions below.
>
> ---
>
> ## Training cost for FLUX and Wan
>
> The learnable sparsity-cost predictor was trained for all models using small conditioning-only datasets (no image pixels): for **FLUX.1-schnell** we used 10,000 captions sampled from COCO, and for **Wan2.1** we used 10,000 captions sampled from WebVid-10M. Training the two-stage sparsity-cost predictor requires roughly **4–10 hours** on **8 AMD MI250 (80GB)** GPUs per experiment. We will make these numbers more prominent in the main text.
>
> ---
>
> ## Reorganization & visualizations
>
> We agree the paper would benefit from a tighter related-work section and moving key figures into the main text. Concretely we will:
>
> - Substantially shorten and focus the Related Work to emphasize the most relevant prior art (feature-caching and token-cache methods).
> - Move the most important qualitative figures (layer-sparsity heatmaps / per-step examples) from the appendix into the main paper to improve accessibility.
> - Keep a compact, curated comparison table of related methods in the appendix for readers who want more detail.
>
> ---
>
> ## Discussion of DyDiT
>
> - **Different acceleration axes.**
>   DyDiT accelerates inference by (i) bypassing token computation when a token is judged unimportant, and (ii) reducing per-layer width (fewer active attention heads / channel groups). In contrast, **DiffSparse** reduces token computation by *reusing cached features*: once a token representation is computed it can be reused across subsequent timesteps rather than recomputed. Because DyDiT changes *how much* each block computes (width / heads) while DiffSparse changes *which* token computations are repeated (temporal reuse), the two approaches are fundamentally **orthogonal** and can be combined for larger speedups.
>
> - **Training cost and sample-aware decisions.**
>   DyDiT determines token importance in a sample-aware manner, which—given the vast diversity of diffusion sample features—incurs a high learning cost; their DiT-XL experiments require on the order of $2\times10^{5}$ fine-tuning iterations. By contrast, DiffSparse first computes token importance using a **training-free compositional attention score**, then learns a compact layer-wise sparsity predictor. This substantially lowers optimization burden: DiffSparse converges in far fewer iterations (on the order of $10^{3}$ iterations for DiT-XL in our experiments), making it easier to train while still producing effective token-selection behavior.
>
> - **Breadth of validation.**
>   DyDiT was validated on DiT-XL only; DiffSparse is evaluated across multiple architectures and tasks (DiT-XL, PixArt-α, FLUX, Wan2.1), covering image and video generation at various resolutions. Given the modest acceleration ratio reported for DyDiT, we did not perform a head-to-head numeric comparison in the current draft, but we will add a focused discussion and (where feasible) empirical comparisons in the revision.
>
> - **Temporal / global optimization.**
>   DyDiT applies a FLOPs-constrained loss primarily at the single-step level, which limits modeling of temporal dependencies across the full sampling trajectory. DiffSparse trains under a global $T$-step objective and uses a two-stage procedure with a dynamic-programming solver to jointly optimize sparsity across timesteps and layers. This enables inter-step and inter-layer coordination so resulting masks better respect the global compute–quality trade-off.
>
> ---
>
> We will add a concise version of the above to the Related Work and Discussion section.

---

> > ### Comment · Reviewer_c1BY · 2025-11-28
> >
> > Thank you for your response! My concerns are well addressed.

---

> > > ### Author Response · Authors · 2025-11-28
> > >
> > > Dear Reviewer c1BY,
> > >
> > > Thank you for your review and for noting our rebuttal addressed your concerns. Your prompt reply is highly encouraging. If you now feel the concerns are resolved, would you consider updating your score to reflect that? We appreciate your time and impartial judgment.
> > >
> > > Best Regards!
> > >
> > > The Authors

---

### Official Review · Reviewer_9MrV · 2025-11-01

**Soundness:** 2
**Presentation:** 3
**Contribution:** 2
**Rating:** 4
**Confidence:** 3

**Summary:**

This paper proposes DiffSparse, a learned token-sparsity allocation framework for accelerating diffusion transformer models.It addresses inefficiencies in existing token caching methods by automating layer-wise sparsity allocation using a differentiable cost predictor and dynamic programming solver.Experiments demonstrate moderate acceleration with limited quality degradation across several diffusion transformer backbones.

**Strengths:**

* The motivation behind the proposed method is well articulated.

* Validates across diverse architectures (DiT, PixArt, FLUX, Wan2.1) and tasks (text-to-image/video, class-conditional), demonstrating robustness.

**Weaknesses:**

* The training process is relatively complex, requiring a bespoke two-stage strategy and 4-10 hours of training on 8 high-end GPUs.

* Results on higher resolution (512×512) are based on one configuration and show only marginal benefit,Table 4 shows the proposed method does not outperform all baselines in FID.

**Questions:**

1.How sensitive is performance to δ and |S|? The limited ablations do not paint a full picture of robustness.

2.Could you include comparisons with state-of-the-art training-free caching approaches such as DiCache and DeepCache.

3.Since the Token Selector computes attention-based ranking signals and dynamically applies binary masks at each step, could the authors quantify the corresponding inference-time overhead and its impact on real wall-clock latency?

4.Given the training cost (4–10 GPU hours with full teacher inference), is the amortized gain meaningful vs. training-free caching or distillation approaches?

---

> ### Author Response · Authors · 2025-11-23
> **Response to Reviewer 9MrV  (part 1)**
>
> Thank you for your detailed and constructive feedback. We appreciate that you found the paper's motivation and cross-architecture validation convincing. Below we respond to each point you raised.
>
> ---
>
> ## Response to Weakness 1 — Training cost
>
> For a standard configuration of PixArt-α, our sparsity cost predictor requires about **4 hours** of training on **8 × AMD MI250** GPUs, versus **≈16 hours** for a genetic-algorithm search baseline. DiffSparse yields better FID while requiring less optimization time in our experiments. Moreover, the learned sparsity predictor is compact (size $(T\times L)\times|S|$) and often transfers from $256\times256$ training to $512\times512$ evaluation, reducing retraining needs for higher resolutions.
>
> By contrast, distillation-based approaches can require substantially larger compute budgets. For example, reported distillation efforts for diffusion models (DMD2) involve $\mathcal{O}(10^3\text{--}10^4)$ GPU·hours (e.g., SD1.5: 1,664 GPU·hr; SDXL: 8,192 GPU·hr), which is orders of magnitude larger than the cost of our method and often depends on private training data that is not publicly available. Importantly, DiffSparse also yields benefits on some distilled models: for instance, on a 4-step distilled model (FLUX.1-schnell) we observe a **1.81×** speedup with no measurable quality drop (see Table 9).
>
> ---
>
> ## Response to Weakness 2
>
> As shown in Table 4, DiffSparse outperforms the existing SOTA method ToCa (ICLR'25) and a 50%-steps-faster sampler by a notable margin. For example, DiffSparse surpasses ToCa by **0.6 FID** (≈2.7% of the original PixArt-α baseline) and surpasses the 50%-steps sampler by **2.63 FID** (≈12.0% of the original PixArt-α baseline).
>
> ---
>
> ## Response to Q1 — Sensitivity to $\delta$ and \(|S|\)
>
> Algorithm 1 uses a warm-start constant $\delta=10$ for the two-stage optimization. Intuitively, $\delta$ injects the Stage-1 prior (the timesteps selected to remain full-step) into Stage-2 by lowering the cost of the “full” candidate at those timesteps. In effect, a larger $\delta$ more strongly encourages preserving full computation at the Stage-1 selected steps.
>
> To quantify this effect we evaluated $\delta\in\{0,5,10,20\}$. Table 1 reports the results on PixArt-α with $T=20$. A moderate warm-start ($\delta=10$) recovers most of the benefit, while $\delta=0$ (no warm-start) removes the Stage-1 prior and yields noticeably worse performance.
>
> We also ablated the candidate-set cardinality $|S|$. Table 2 summarizes experiments for different sparse-interval granularities: interval $0.25$ (i.e., $|S|=5$, candidate set $\{0,0.25,0.5,0.75,1.0\}$) provides the best trade-off between discretization granularity and training stability in our tests. We additionally evaluated $|S|\in\{2,11\}$ (intervals 1.0 and 0.1) and found that extreme coarseness or extreme fineness harms either final quality or stability; $|S|=5$ remains the most practical choice.
>
> **Table 1 — Ablation of warm-start strength $\delta$ (PixArt-α, $T=20$). Lower FID and higher CLIP are better.**
>
> | δ | FID ↓ | CLIP ↑ |
> |---:|---:|---:|
> | 0  | 27.40 | 0.163 |
> | 5  | 27.01 | 0.164 |
> | 10 | 26.91 | 0.164 |
> | 20 | 26.95 | 0.164 |
>
> **Table 2 — Ablation over sparse-interval granularity (PixArt-α, $T=20$).**
>
> | Interval | \|S\| | FID ↓ | CLIP ↑ |
> |---:|---:|---:|---:|
> | 0.10  | 11 | 27.96 | 0.163 |
> | 0.125 | 9  | 27.91 | 0.163 |
> | 0.25  | 5  | **26.91** | 0.164 |
> | 0.50  | 3  | 27.54 | 0.164 |
> | 1.00  | 2  | 28.22 | 0.162 |
>
> We will include these additional experiments in the revised manuscript.
>
> ---
>
> ## Response to Q2 — Comparison with training-free caching approaches (DiCache, DeepCache)
>
> We have already compared against several state-of-the-art methods such as ToCa (ICLR'25) and TaylorSeer (ICCV'25). In response to your suggestion, we added dedicated comparisons to training-free caching approaches (DeepCache (CVPR'24) and DiCache (arXiv'25)). Results on MS-COCO2017 with PixArt-α and $T=20$ DPM++ steps are shown in Table 3. DiffSparse demonstrates a clear quality–speed advantage over these training-free baselines in this setting.
>
> **Table 3 — Text-to-image generation on MS-COCO2017 with PixArt-α and 20 DPM++ steps. “MACs (T)” reports the model compute at $T$ steps; Speedup is relative to the full model.**
>
> | Method | MACs (T) ↓ | Speedup ↑ | FID-30k ↓ | CLIP ↑ |
> |---|---:|---:|---:|---:|
> | PixArt-α (full) | 2.86 | 1.00× | 28.20 | 0.163 |
> | DeepCache (𝒩=2) | 1.48 | 1.61× | 29.61 | 0.163 |
> | DiCache | 1.63 | 1.77× | 28.19 | 0.164 |
> | **DiffSparse (R = 43%)** | **1.64** | **1.74×** | **26.91** | **0.164** |
>
> ---

---

> > ### Comment · Reviewer_9MrV · 2025-11-24
> >
> > Thanks for the rebuttal. My concerns have been solved. I raised the score to 6. good luck

---

> > > ### Author Response · Authors · 2025-11-24
> > > **Thanks for Raising the Score**
> > >
> > > We greatly appreciate your valuable feedback and the improved scores. Your recognition is highly encouraging.
> > >
> > > Best Regards!

---

> ### Author Response · Authors · 2025-11-23
> **Response to Reviewer 9MrV (part 2)**
>
> ## Response to Q3 — Inference-time overhead of the Token Selector
>
> The DP solver and the two-stage optimization are **training-time** procedures: Stage 1 solves for $C_f$ via DP and Stage 2 fine-tunes $C_l$. The DP solver runs in approximately **≈30 seconds** for the configurations reported, but it is **not** executed at inference time. At inference, the model only uses the precomputed masks; the token scoring, top-$K$ selection, and cache operations are lightweight compared to attention and MLP FLOPs. The real wall-clock inference speedups are presented in Tables 1–3 of the manuscript.
>
> ---
>
> ## Response to Q4 — Is the amortized gain meaningful?
>
> For our main configurations we report **≈4 hours** of training for the differentiable optimization, versus **≈16 hours** for a genetic-algorithm search baseline. DiffSparse yields better FID while requiring less optimization time in our experiments. Moreover, the learned sparsity predictor is compact (size $(T\times L)\times|S|$) and often transfers from $256\times256$ training to $512\times512$ evaluation, reducing retraining needs for higher resolutions.
>
> By contrast, distillation-based approaches can require substantially larger compute budgets. For example, reported distillation efforts for diffusion models (DMD2) involve $\mathcal{O}(10^3\text{--}10^4)$ GPU·hours (e.g., SD1.5: 1,664 GPU·hr; SDXL: 8,192 GPU·hr), which is orders of magnitude larger than the cost of our method and often depends on private training data that is not publicly available. Importantly, DiffSparse also yields benefits on some distilled models: for instance, on a 4-step distilled model (FLUX.1-schnell) we observe a **1.81×** speedup with no measurable quality drop (see Table 9).
>
> ---
>
> We appreciate the reviewer’s suggestions. The additional clarifications and experiments described above will be included in the revised manuscript.

---

### Official Review · Reviewer_ZHnY · 2025-11-01

**Soundness:** 3
**Presentation:** 2
**Contribution:** 2
**Rating:** 4
**Confidence:** 2

**Summary:**

DiffSparse is a novel framework for accelerating Diffusion Transformer models through learnable token-level sparsity optimization. Unlike existing token caching methods that require manual sparsity configuration and depend on predefined full-step computations, DiffSparse automatically learns optimal layer-wise sparsity allocation in an end-to-end manner while eliminating the need for complete forward passes at certain steps.

**Strengths:**

The method comprises three key components: (1) a Token Selector that computes importance scores based on self-attention influence, cross-attention entropy, cache reuse frequency, and spatial distribution to determine which tokens should be recomputed versus reused from cache; (2) a Learnable Sparsity Cost Predictor with trainable parameters that outputs a normalized cost matrix quantifying the quality impact of different sparsity rates for each layer at each timestep; and (3) a Dynamic Programming Solver that finds the optimal sparsity configuration across all layers and timesteps by minimizing cumulative cost while satisfying a global sparsity constraint , with gradients flowing through discrete decisions via Straight-Through Estimation.

**Weaknesses:**

1. Symbol M is overloaded with two conflicting meanings within the same section.  First, M (introduced in Section 3.2, "Learnable Sparsity Cost Predictor") represents a binary mask matrix for token selection at each layer and timestep,  with dimension equal to the number of tokens N. Second, M appears as the  summation upper bound in Equation 5, representing the count of importance  criteria combined in the token ranking score. These two usages denote entirely  different mathematical objects—a vector versus a scalar integer—creating  potential confusion for readers.
2.Compared to "training-free" competing methods (such as ToCa), this approach  requires additional training but does not demonstrate a clear advantage in  acceleration performance.

**Questions:**

1. Does the method require retraining for each new model, new resolution, or  new sampling step count? Is this training overhead worthwhile?
2.While Table 5 demonstrates that the Attention strategy outperforms other  strategies overall, it fails to analyze the individual contributions of
the four components (self-attention influence, cross-attention focus, cache-reuse frequency, etc)  within the Attention method .

---

> ### Author Response · Authors · 2025-11-23
> **Response to Reviewer ZHnY**
>
> Thank you for the careful reading and constructive comments. We address the concerns below.
>
> ---
>
> ## Notation clash (symbol $M$)
>
> You are correct: the manuscript currently uses the same letter $M$ for two different mathematical objects (a binary per-layer / per-timestep mask and the number of score components in Eq. (5)).
> - The mask is defined in the *Learnable Sparsity Cost Predictor* section as a binary token-selection mask $\mathbf{m}\in\{0,1\}^N$.
> - The token-score formula (Eq. (5)) uses $M$ as the number of importance criteria summed in the ranking score.
>
> **Planned fix.** We will rename the summation bound in Eq. (5) to $Q$ (the number of scoring criteria) and $Q$ denotes the number of criteria (self-attention influence, cross-attention focus, cache-reuse frequency, spatial bonus). This small notation change will remove reader confusion; the underlying algorithm and results are unchanged.
>
> ---
>
> ## Additional training vs. training-free methods (e.g., ToCa)
>
> - **Empirical comparison.** Although DiffSparse requires learning a compact cost predictor of size $(T\times L)\times|S|$, our experiments show it yields a superior quality–speed trade-off compared to several baselines, including ToCa and other token-caching methods. For example, on PixArt-$\alpha$ with $T=20$ sampling steps, DiffSparse achieves FID $26.91$ at approximately $1.74\times$ speedup, which is a $+4.6$% improvement relative to the full baseline. By contrast, ToCa attains FID $28.35$ at a comparable acceleration (a $-0.5$% change relative to the same baseline). This corresponds to a $5.1$% relative improvement in FID of DiffSparse over ToCa (see main results in Table 1).
>
> - **Training cost and practicality.** Training the sparsity cost predictor is lightweight in practice: we train on small conditioning-only datasets (10k captions / labels without images) and report training runs of about **4–10 hours** on 8 × AMD MI250 GPUs. This modest one-time offline cost yields a learned sparse policy that (a) transfers across image resolutions (Table 4), and (b) outperforms simple search-based approaches while being faster to obtain.
>
> ---
>
> ## Response to Q1 — Retraining, transferability, and whether the overhead is worthwhile
>
> - **Retraining requirement.** If you change the *model architecture* significantly (e.g., different number of layers $L$ or different block structure), retraining or at least fine-tuning is required when the *temporal* or *architectural* axes change ($T$ or $L$) because the predictor’s parameters are tied to $(T,L,|S|)$. By contrast, changing only the token length $N$ (image resolution) usually **does not** require full retraining: the predictor's size and learned entries are independent of $N$, and we show that a predictor trained at $256\times256$ transfers to $512\times512$ (Table 4).
>
> - **Why the training is worthwhile.** The learned cost predictor enables automatic per-layer, per-timestep sparsity allocation (no hand-crafted schedules), which leads to both higher acceleration and better sample quality compared to training-free alternatives (see Tables 1-4 and the image/video results). Because the predictor depends on $(T,L,|S|)$ and **not** on sequence length $N$, it is memory- and compute-efficient to deploy. The predictor trained at lower resolution transfers to higher resolutions, reducing retraining needs when only image size changes.
>
> ---
>
> ## Response to Q2 — Component ablations of the attention-based score
>
> Table 5 compares three importance metrics (attention-based score, cosine similarity, $\ell_2$ norm) and shows the attention-based score performs best overall. We also performed a component-wise ablation by removing one term at a time from the composite attention score ($-s_1$, $-s_2$, $-s_3$, $-B$) to measure each term’s contribution.
>
> **Ablation results (FID on PixArt-$\alpha$):**
>
> | Variant | Removed component | FID ↓ | CLIP ↑ |
> |---|---|---|---:|
> | DiffSparse (full) | — | 26.91 | 0.164|
> | −s₁ | self-attention influence | 27.11 | 0.164|
> | −s₂ | cross-attention focus | 27.48 | 0.163|
> | −s₃ | cache-reuse frequency | 27.23 | 0.164|
> | −B  | spatial bonus | 27.05 | 0.164|
>
> We expect these ablations to strengthen the empirical story. We will include them in the revised paper.

---

> > ### Comment · Reviewer_ZHnY · 2025-11-27
> >
> > Thank you for the clarifications. I have other two concerns.
> > (1) Regarding the FID of ToCa, i note an inconsistency: Table 1 reports an FID of 28.35, whereas the value you provided is 28.38. I kindly request clarification on which value represents the actual experimental result, as this affects the accuracy of the claimed performance improvements.
> > (2) In Table 3, you compared DiffSparse with the Wan2.1-1.3B model and showed that DiffSparse achieves the best VBench score while providing 2.05× speedup. However, I have a question about the experimental setup: Does this phenomenon persist when using different numbers of inference timesteps (T) for the Wan2.1 model?

---

> > > ### Author Response · Authors · 2025-11-27
> > > **Author Response and Clarification**
> > >
> > > Thanks for the careful reading and helpful questions. Below are our responses.
> > >
> > > **Response to (1).** The correct FID for ToCa is **28.35**; the value **28.38** was a minor reporting discrepancy in an earlier comment and has been corrected. Using 28.35 as the reference, DiffSparse yields a **5.1%** relative improvement in FID over ToCa, where the relative improvement is computed as (FID_ToCa - FID_DiffSparse) / FID_PixArt_Baseline * 100%
> > >
> > >
> > > **Response to (2).** We validated DiffSparse across multiple inference-step regimes (50 steps for DiT-XL; 20 steps for PixArt-$\alpha$ and Wan2.1; 4 steps for FLUX), and the quality–speed advantages persist across these settings. Table 3 reports the Wan2.1 comparison at 20 steps (the configuration used for that experiment); the reported 2.05$\times$ speedup with the best VBench is measured against baselines using the same number of steps. By contrast, naively reducing the number of sampling steps in the baseline typically causes a noticeable drop in generation quality (see Tables 1, 2, and 4), which shows that DiffSparse preserves fidelity more effectively than simple step reduction. If helpful, we can add additional Wan2.1 baselines at other step values in the revision.
> > >
> > > We hope this addresses your questions. If you have any further concerns, please do not hesitate to let us know.

---

> > > > ### Author Response · Authors · 2025-11-27
> > > > **Author Response**
> > > >
> > > > We evaluated an additional Wan2.1 baseline with 50% fewer sampling steps (10 steps) and obtained a VBench score of **43.14**, which is lower than DiffSparse's **43.83** at a comparable acceleration ratio. These experiments have been added to the revised manuscript. We hope this addresses your concern. Please let us know if you have any further questions.

---

> ### Author Response · Authors · 2025-11-28
> **Kindly Request for Your Feedback!**
>
> Dear Reviewer ZHnY,
>
> Thank you for your careful review. Below is a short summary of our rebuttal and new results — we would very much appreciate your feedback on whether these address your concerns.
>
> **Summary of our responses**
>
> 1. **Notation fix**
>
>    * We will rename the summation bound in Eq.(5) from $M$ to $Q$ (number of scoring criteria) to avoid clash with the binary mask $m$. This is a notational fix only, our algorithms and results are unchanged.
>
> 2. **Quality vs. training-free methods (key numbers)**
>
>    * DiffSparse (PixArt-α, T=20): FID = **26.91**, ≈ **1.74×** speedup (≈ +4.6% vs baseline).
>    * ToCa (same setup): FID = **28.35** (worse quality at comparable acceleration).
>    * This is a **~5.1% relative FID improvement** of DiffSparse over ToCa (Table 1).
>
> 3. **Training cost & practicality**
>
>    * Training the compact cost predictor is lightweight: **~4–10 hours** on **8 × AMD MI250** (one-time training cost).
>    * Predictor size = $(T\times L)\times|S|$ (independent of token length $N$), and a predictor trained at $256×256$ transfers to $512×512$ (Table 4), so in-practice retraining is limited to changes in $T$ or $L$, not image resolution.
>
> 4. **Why retrain / why worthwhile**
>
>    * Learned per-layer, per-timestep allocation removes hand-crafted scheduling, yielding better quality–speed trade-offs than training-free approaches and simple step-reduction. Predictor is memory- and compute-efficient to deploy.
>
> 5. **Ablations & robustness**
>
>    * Attention-based composite scoring outperforms cosine and ℓ₂ metrics (Table 5). Component-wise removals increase FID modestly (PixArt-α results below):
>
> | Variant           |        Removed component |       FID |  CLIP |
> | ----------------- | -----------------------: | --------: | ----: |
> | DiffSparse (full) |                        — | 26.91 | 0.164 |
> | −s₁               | self-attention influence |     27.11 | 0.164 |
> | −s₂               |    cross-attention focus |     27.48 | 0.163 |
> | −s₃               |    cache-reuse frequency |     27.23 | 0.164 |
> | −B                |            spatial bonus |     27.05 | 0.164 |
>
> 6. **Multi-regime validation**
>
>    * Evaluated across many samplers / step regimes (DiT-XL at 50 steps; PixArt-α and Wan2.1 at 20 steps; FLUX at 4 steps). Example: Wan2.1 achieves **2.05×** speedup (measured at 20 steps, VBench comparisons in Table 3). A 10-step Wan2.1 baseline yields VBench **43.14** vs DiffSparse **43.83** at comparable acceleration, showing DiffSparse preserves fidelity better than naive step reduction.
>
> ---
>
> We hope these clarifications (notation fix, empirical numbers, training cost, ablations, and multi-regime validation) address your concerns. If so, we would be grateful if you would consider revising your rating. If you have any further questions regarding our responses, we would be happy to provide additional clarification.
> Thank you for your time and consideration.
>
> Sincerely,
>
> The Authors

---

### Meta-Review · Area_Chair_Xyb2 · 2026-01-05

**Summary:**

c1BY were initially positive, and questions raised are well addressed by the authors in the rebuttal. 9MrV was initially slightly negative with concerns including training complexity, marginal improvements in Tab4, parameter sensitivity, and comparisons to training free caching based approaches, and authors have addressed all the concerns, and the reviewer confirmed the concerns are addressed well. ZHnY raised concerns on the overloaded notation, and lack of performance advantage over ToCa, and lack of ablations, and authors have provided reasonable responses. The reviewer asked further questions, for which authors seem also addressed with clarifications. wjJ7 raised several concerns and questions including insufficient overhead analysis and ablation, effectiveness on few-step inferences, etc.

**Reviewer Concerns:**

c1BY only had some minor questions and suggestions to reorganization of writing, which are addressed in the rebuttal.
9MrV's concerns on the complexity, marginal improvements, and parameter sensitivity are addressed in the rebuttal (confirmed by the reviewer).
ZHnY's concerns on overloaded notation, lack of performance improvement and ablations seem addressed with detailed rebuttal.
wjJ7's concern on insufficent overhead analysis and ablations seem partially addressed in the rebuttal.

**Reviewer Scores:**

9MrV probably will increase scores to 6.
c1BY probably will stay with 6.
ZHnY probably will increase scores to 6.
wjJ7: have many concerns, and may have stayed at 4.

---

### Decision · Program_Chairs · 2026-01-26

Accept (Poster)